# ONTOPROTEIN: PROTEIN PRETRAINING WITH GENE ONTOLOGY EMBEDDING

**Ningyu Zhang**[1,2,3*]   **Zhen Bi**[2,3*]   **Xiaozhuan Liang**[2,3*]   **Siyuan Cheng**[2,3*]   **Haosen Hong**[4]
**Shumin Deng**[1,3]   **Qiang Zhang**[1,4]   **Jiazhang Lian**[4]   **Huajun Chen**[1,3,4†]

[1]College of Computer Science and Technology, Zhejiang University
[2]School of Software Technology, Zhejiang University
[3]Alibaba-Zhejiang University Joint Research Institute of Frontier Technologies
[4]Hangzhou Innovation Center, Zhejiang University
{zhangningyu,bizhen_zju,liangxiaozhuan,22151070}@zju.edu.cn
{231sm,12028071,jzlian,qiang.zhang.cs,huajunsir}@zju.edu.cn

## ABSTRACT

Self-supervised protein language models have proved their effectiveness in learning the proteins representations. With the increasing computational power, current protein language models pre-trained with millions of diverse sequences can advance the parameter scale from million-level to billion-level and achieve remarkable improvement. However, those prevailing approaches rarely consider incorporating knowledge graphs (KGs), which can provide rich structured knowledge facts for better protein representations. We argue that informative biology knowledge in KGs can enhance protein representation with external knowledge. In this work, we propose **OntoProtein**, the first general framework that makes use of structure in GO (Gene Ontology) into protein pre-training models. We construct a novel large-scale knowledge graph that consists of GO and its related proteins, and gene annotation texts or protein sequences describe all nodes in the graph. We propose novel contrastive learning with knowledge-aware negative sampling to jointly optimize the knowledge graph and protein embedding during pre-training. Experimental results show that OntoProtein can surpass state-of-the-art methods with pre-trained protein language models in TAPE benchmark and yield better performance compared with baselines in protein-protein interaction and protein function prediction[1].

## 1 INTRODUCTION

Protein science, the fundamental macromolecules governing biology and life itself, has led to remarkable advances in understanding the disease therapies and human health (Vig et al. (2021)). As a sequence of amino acids, protein can be viewed precisely as a language, indicating that they may be modeled using neural networks that have been developed for natural language processing (NLP). Recent self-supervised pre-trained protein language models (PLMs) such as ESM (Rao et al. (2021b)), ProteinBERT (Brandes et al. (2021)), ProtTrans (Elnaggar et al. (2020)) which can learn powerful protein representations, have achieved promising results in understanding the structure and functionality of the protein. Yet existing PLMs for protein representation learning generally cannot sufficiently capture the biology factual knowledge, which is crucial for many protein tasks but is usually sparse and has diverse and complex forms in sequence.

By contrast, knowledge graphs (KGs) from gene ontology[2] contain extensive biology structural facts, and knowledge embedding (KE) approaches (Bordes et al. (2013), Zheng et al. (2021)) can efficiently embed them into continuous vectors of entities and relations. For example, as shown in Figure 1, without knowing *PEX5* has specific biological processes and cellular components, it

---

[*]Equal contribution and shared co-first authorship.
[†]Corresponding author.

[1]Code and datasets are available in `https://github.com/zjunlp/OntoProtein`.
[2]`http://geneontology.org/`

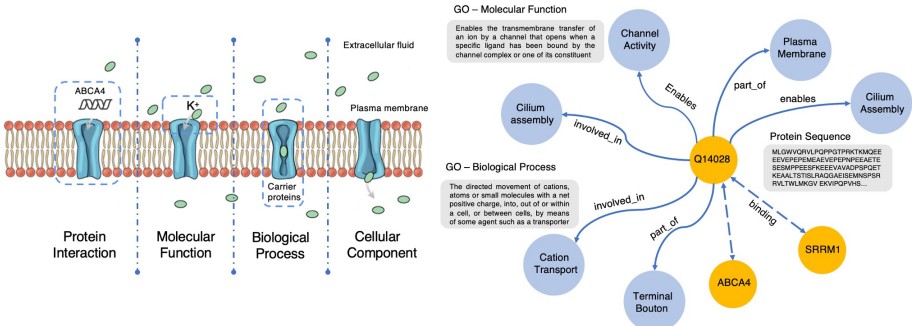

Figure 1: **Left**: A protein example with biology knowledge (molecular function, biological process and cellular component): $K^+$ (potassium ion) Cyclic nucleotide-gated cation channel protein. **Right**: The corresponding sub-graph regarding $K^+$ carrier proteins in **ProteinKG25**. **Yellow** nodes are protein sequences and **blue** nodes are GO (Gene Ontology) entities with biological descriptions.

is challenging to recognize its interaction with other proteins. Furthermore, since *protein's shape determines its function*, it is more convenient for models to identify protein's functions with the prior knowledge of protein functions having similar shapes. Hence, considering rich knowledge can lead to better protein representation and benefits various biology applications, e.g., protein contact prediction, protein function prediction, and protein-protein interaction prediction. However, different from knowledge-enhanced approaches in NLP (Zhang et al. (2019b), Wang et al. (2021b), Wang et al. (2021a)) , protein sequence and gene ontology are two different types of data. Note that protein sequence is composed of amino acids while gene ontology is a knowledge graph with text description; thus, severe issues of structured knowledge encoding and heterogeneous information fusion remain.

In this paper, we take the first to propose protein pre-training with gene ontology embedding (**OntoProtein**), which is the first general framework to integrate external knowledge graphs into protein pre-training. We propose a hybrid encoder to represent language text and protein sequence and introduce contrastive learning with knowledge-aware negative sampling to jointly optimize the knowledge graph and the protein sequence embedding during pre-training. For the KE objective, we encode the node descriptions (go annotations) as their corresponding entity embeddings and then optimize them following vanilla KE approaches (Bordes et al. (2013)). We further leverage gene ontology of molecular function, cellular component, and biological process and introduce a knowledge-aware negative sampling method for the KE objective. For the MLM (Mask Language Modeling) objective, we follow the approach of existing protein pre-training approaches (Rao et al. (2021b)). OntoProtein has the following strengths:

(1) OntoProtein inherits the strong ability of protein understanding from PLMs with the MLM object. (2) OntoProtein can integrate biology knowledge into protein representation with the supervision from KG by the KE object. (3) OntoProtein constitutes a model-agnostic method and is readily pluggable into a wide range of protein tasks without additional inference overhead since we do not modify model architecture but add new training objectives.

For pre-training and evaluating OntoProtein, we need a knowledge graph with large-scale biology knowledge facts aligned with protein sequences. Therefore, we construct **ProteinKG25**, which contains about 612,483 entities, 4,990,097 triples, and aligned node descriptions from GO annotations. To the best of our knowledge, it is the first large-scale KG dataset to facilitate protein pre-training. We deliver data splits for both the inductive and the transductive settings to promote future research.

To summarize, our contribution is three-fold: (1) We propose OntoProtein, the first knowledge-enhanced protein pre-training approach that brings promising improvements to a wide range of protein tasks. (2) By contrastive learning with knowledge-aware sampling to jointly optimize knowledge and protein embedding, OntoProtein shows its effectiveness in widespread downstream tasks, including protein function prediction, protein-protein interaction prediction, contact prediction, and so on. (3) We construct and release the ProteinKG25, a novel large-scale KG dataset, promoting the

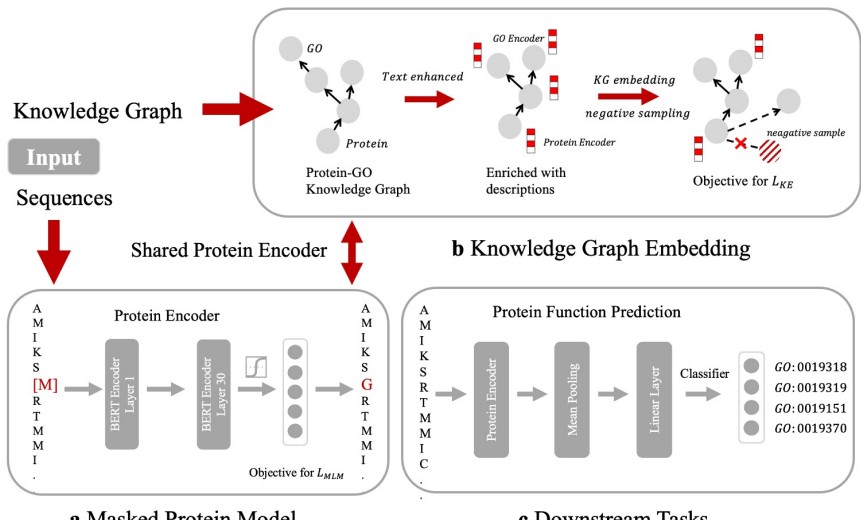

Figure 2: Overview of our proposed OntoProtein, which jointly optimize knowledge graph embedding and masked protein model (Best viewed in color.).

research on protein language pre-training. (4) We conduct extensive experiments in widespread protein tasks, including TAPE benchmark, protein-protein interaction prediction, and protein function prediction, which demonstrate the effectiveness of our proposed approach.

## 2 METHODOLOGIES

We begin to introduce our approach of protein pre-training with ontology embedding (OntoProtein), as shown in Figure 2. OntoProtein incorporates external knowledge from Gene Ontology (Go) into language representations by jointly optimizing two objectives. We will first introduce the hybrid encoder, masked protein modeling, and knowledge encoder, and then we will present the details of contrastive learning with knowledge-aware negative sampling. Finally, we will illustrate the overall pre-training objects.

### 2.1 HYBRID ENCODER

We first introduce the hybrid encoder to represent protein and GO knowledge. For the protein encoder, we use the pre-trained ProtBert from Elnaggar et al. (2020). ProtBert is pre-trained using the BERT architecture with UniRef100 datasets. Compared to BERT Devlin et al. (2019), ProtBert encodes amino acid sequences into token level or sentence level representations, which can be used for downstream protein tasks such as contacts prediction tasks. The encoder takes a protein sequence of $N$ tokens $(x_1, ..., x_N)$ as inputs, and computes contextualized amnio acid representation $H^i_{Protein}$ and sequence representation $H_{Protein}$ via *mean pooling*. To bridge the gap between text and protein, we utilize affine transformation (an extra linear layer) to project those representation to the same space. We will discuss details of learning protein representation in Section Mask Protein Modeling.

For the Go encoder, we leverage BERT (Devlin et al. (2019)), a Transformer (Vaswani et al. (2017)) based text encoder for biological descriptions in Gene Ontology entities. Specifically, we utilize the pre-trained language model from (Gu et al. (2020))[3]. The encoder takes a sequence of $N$ tokens $(x_1, ..., x_N)$ as inputs, and computes Go representations $H_{GO} \in R^{N \times d}$ by averaging all the token embeddings.

---

[3]https://huggingface.co/microsoft/BiomedNLP-PubMedBERT-base-uncased-abstract-fulltext

Since the relations in Gene Ontology are important for representing the knowledge of biology features, thus, we utilize a relation encoder with the random initialization, and those embeddings of relations will be optimized and updated during pre-training.

## 2.2 KNOWLEDGE EMBEDDING

We leverage the knowledge embedding (KE) objective to obtain representations in the pre-training process since Gene Ontology is actually a factual knowledge graph. Similar to Bordes et al. (2013), we use distributed representations to encode entities and relations. The knowledge graph here consists of lots of triples to describe relational facts. We define a triplet as $(h, r, t)$, where $h$ and $t$ are head and tail entities, $r$ is the relation whose type usually is pre-defined in the schema[4]. Note that there are **two different types of nodes** $e_{GO}$ and $e_{protein}$ in our knowledge graph. $e_{GO}$ is denoted as nodes that exist in the gene ontology, such as molecular function or cellular component nodes, and $e_{GO}$ can be described by annotation texts. $e_{protein}$ is the protein node that links to the gene ontology, and we also represent $e_{protein}$ with amnio acids sequences. Concretely, the triplets in this knowledge graph can be divided into two groups, $triple_{GO2GO}$ and $triple_{Protein2GO}$. To integrate multi-modal descriptions into the same semantic space and address the heterogeneous information fusion issue, we utilize hybrid encoders introduced in the previous Section. Note that protein encoder and GO encoder represent protein sequence and GO annotations separately.

## 2.3 MASKED PROTEIN MODELING

We use masked protein modeling to optimize protein representations. The masked protein modeling is similar to masked language modeling (MLM). During model pre-training, we use a 15% probability to mask each token (amino acid) and leverage a cross-entropy loss $\ell_{MLM}$ to estimate these masked tokens. We initialize our model with the pre-trained model of ProtBert and regard $\ell_{MLM}$ as one of the overall objectives of OntoProtein by jointly training KE (knowledge embedding) and MLM. Our approach is model-agnostic, and other pre-trained models can also be leveraged.

## 2.4 CONTRASTIVE LEARNING WITH KNOWLEDGE-AWARE NEGATIVE SAMPLING

Knowledge embedding (KE) is to learn low-dimensional representations for entities and relations, and contrastive estimation represents a scalable and effective method for inferring connectivity patterns. Note that a crucial aspect of contrastive learning approaches is the choice of corruption distribution that generates hard negative samples, which force the embedding model to learn discriminative representations and find critical characteristics of observed data. However, previous approaches either employ too simple corruption distributions, i.e., uniform, yielding easy uninformative negatives, or sophisticated adversarial distributions with challenging optimization schemes. Thus, in this paper, we propose contrastive learning with knowledge-aware negative sampling, an inexpensive negative sampling strategy that utilizes the rich GO knowledge to sample negative samples. Formally, the KE objective can be defined as:

$$\ell_{KE} = -\log \sigma(\gamma - d(h, t)) - \sum_{i=1}^{n} \frac{1}{n} \log \sigma(d(h_i', t_i') - \gamma) \tag{1}$$

$(h_i', t_i')$ is the negative sample, in which head or tail entities are random sampled to construct the corrupt triples. $n$ is the number of negative samples, $\sigma$ is the sigmoid function, and $\gamma$ means the margin. $d$ is the scoring function, and we use TransE (Bordes et al. (2013)) for simplicity, where

$$d_r(h, t) = \|h + r - t\| \tag{2}$$

Specifically, we define triple sets and entity sets as $T$ and $E$, all triplets are divided into two groups. If the head entity is protein node and the tail entity is GO node, we denote the triple as $T_{protein-GO}$. Similarly, if head and tail entities are both GO nodes, we denote them as $T_{GO-GO}$. As Gene Ontology describes the knowledge of the biological domain concerning three aspects, all entities in Gene Ontology belong to MFO (Molecular Function), CCO (Cellular Component), or BPO (Biological Process).

---

[4]The schema of the knowledge graph can be found in Appendix A.1

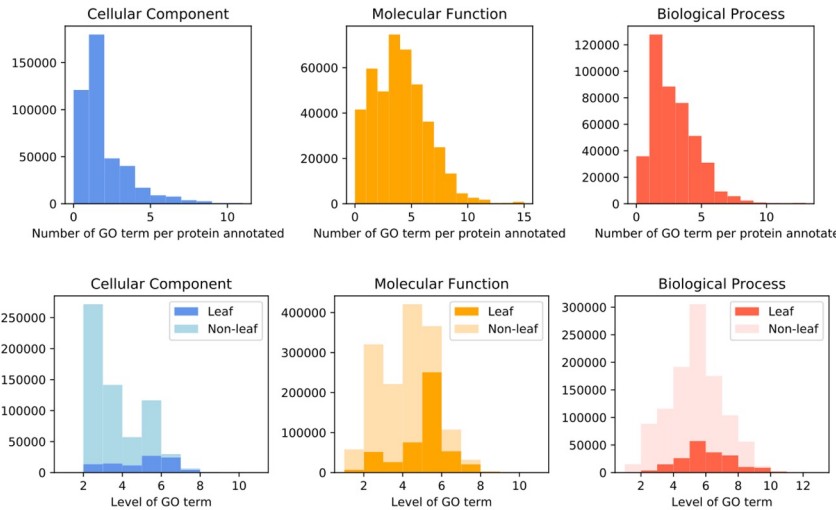

Figure 3: **Top**: Data Distribution of GO Terms. **Bottom**: Statistics of Protein-GO Term.

To avoid plain negative samples, for those $T_{GO-GO}$ triples, we sample triples by replacing entities with the same aspect (MFO, CCO, BPO)[5]. Finally, we define the negative triple sets $T'$ and positive triple as $(h, r, t)$, the negative sampling process can be described as follows:

$$T'_{GO-GO(h,r,t)} = \{(h', r, t) \mid h' \in E', h \in E'\} \cup \{(h, r, t') \mid t' \in E', t \in E'\}$$
$$T'_{Protein-GO(h,r,t)} = \{(h, r, t') \mid t' \in E'\} \tag{3}$$

where $E' \in \{E_{MFO}, E_{CCO}, E_{BPO}\}$, and we only replace the tail entities for $T_{Protein-GO}$ triples.

## 2.5 Pre-training Objective

We adopt the mask protein modeling object and knowledge embedding objective to construct the overall object of the OntoProtein. We jointly optimize the overall object as follows:

$$\ell = \alpha \ell_{KE} + \ell_{MLM} \tag{4}$$

where $\alpha$ is the hyper-parameter. Our approach can be embedded into existing fine-tuning scenarios.

## 3 Experiment

Extensive experiments have been conducted to prove the effectiveness of our approach. In the pre-training stage, we construct a new knowledge graph dataset that consists of Gene Ontology and public annotated proteins. Our proposed model is pre-trained with this dataset and evaluated in several downstream tasks. We evaluate OntoProtein in protein function prediction, protein-protein interaction and TAPE benchmark (Rao et al. (2019)).

### 3.1 Datasets

**Pre-training Dataset** To incorporate Gene Ontology knowledge into language models, we build a new pre-training dataset called ProteinKG25[6], which is a large-scale KG dataset with aligned descriptions and protein sequences respectively to GO terms[7] and proteins entities. Gene Ontology

---

[5]For $T_{protein-GO}$ triples, it is also intuitive to replace the proteins with their homologous proteins to generate hard negative triples, and we leave this for future works.

[6]https://zjunlp.github.io/project/ProteinKG25/

[7]The structure of GO can be described in terms of a graph, where each GO term is a node, and the relationships between the terms are edges between the nodes.

| Method | Structure | | | Evolutionary | Engineering | |
| --- | --- | --- | --- | --- | --- | --- |
| | SS-Q3 | SS-Q8 | Contact | Homology | Fluorescene | Stability |
| LSTM | 0.75 | 0.59 | 0.26 | 0.26 | 0.67 | 0.69 |
| TAPE Transformer | 0.73 | 0.59 | 0.25 | 0.21 | **0.68** | 0.73 |
| ResNet | 0.75 | 0.58 | 0.25 | 0.17 | 0.21 | 0.73 |
| MSA Transformer | - | **0.73** | **0.49** | - | - | - |
| ProtBert | 0.81 | 0.67 | 0.35 | **0.29** | 0.61 | **0.82** |
| OntoProtein | **0.82** | 0.68 | 0.40 | 0.24 | 0.66 | 0.75 |

Table 1: Results on TAPE Benchmark. SS is a secondary structure task that evaluates in CB513. In contact prediction, we test medium- and long-range using P@L/2 metrics. In protein engineering tasks, we test fluorescence and stability prediction using spearman's $\rho$ metric.

consists of a set of GO terms (or concepts) with relations that operate between them, e.g., molecular function terms describe activities that occur at the molecular level. A GO annotation is a statement about the function of a particular gene or gene product, e.g., the gene product "cytochrome c" can be described by the molecular function oxidoreductase activity. Due to the connection between Gene Ontology and Gene Annotations, we combine the two structures into a unified knowledge graph. For each GO term in Gene Ontology, we align it to its corresponding name and description and concatenate them by a colon as an entire description. For each protein in Gene annotation, we align it to the Swiss-Prot[8], a protein knowledge database, and extract its corresponding sequence as its description. In ProteinKG25, there exists 4,990,097 triples, including 4,879,951 $T_{protein-GO}$ and 110,146 $T_{GO-GO}$ triples. Figure 3 illustrate the statistics of our ProteinKG25. Detailed construction procedure and analysis of pre-train datasets can be found in Appendix A.1.

**Downstream Task Dataset** We use TAPE as the benchmark (Rao et al. (2019)) to evaluate protein representation learning. There are three types of tasks in TAPE, including structure, evolutionary, and engineering for proteins. Following Rao et al. (2021a), we select 6 representative datasets including secondary structure (SS), contact prediction to evaluate OntoProtein.

Protein-protein interactions (PPI) are physical contacts of high specificity established between two or more protein molecules; we regard PPI as a sequence classification task and use three datasets with different sizes for evaluation. STRING is built by Lv et al. (2021), which contains 15,335 proteins and 593,397 PPIs. We also use SHS27k and SHS148k, which are generated by Chen et al. (2019).

Protein function prediction aims to assign biological or biochemical roles to proteins, and we also regard this task as a sequence classification task. We build a new evaluation dataset based on our ProteinKG25 following the standard CAFA protocol (Zhou et al. (2019)). Specifically, we design two evaluation settings, the transductive setting and the inductive setting, which simulate two scenarios of gene annotation in reality. In the transductive setting, the model can generate embeddings of unseen protein entities with entity descriptions. On the contrary, for the inductive setting, those entities have occurred in the pre-training stage. The detailed construction of the dataset can be found in Appendix A.1. As shown in Figure 3, proteins are, on average, annotated by 2 terms in CCO, 4 in MFO, and 3 in BPO, indicating that protein function prediction can be viewed as a multi-label problem. Notably, we notice that leaf GO terms tend to have more specific concepts than non-leaf GO terms. Meanwhile, there exists a challenging long-tail issue for the function prediction task.

## 3.2 RESULTS

### TAPE BENCHMARK

**Baselines** In TAPE, we evaluate our OntoProtein compared with five baselines. The first is the model with LSTM encoding of the input amino acid sequence, which provides a simple baseline. The second is TAPE Transformer that provides a basic transformer baseline. We further select ResNet from He et al. (2016) as a baseline. The forth is the MSA Transformer (Rao et al. (2021a)).

---

[8]https://www.uniprot.org/

| Methods | SHS27k BFS | SHS27k DFS | SHS148k BFS | SHS148k DFS | STRING BFS | STRING DFS |
|---|---|---|---|---|---|---|
| DPPI | 41.43 | 46.12 | 52.12 | 52.03 | 56.68 | 66.82 |
| DNN-PPI | 48.09 | 54.34 | 57.40 | 58.42 | 53.05 | 64.94 |
| PIPR | 44.48 | 57.80 | 61.83 | 63.98 | 55.65 | 67.45 |
| GNN-PPI | 63.81 | 74.72 | 71.37 | 82.67 | 78.37 | 91.07 |
| GNN-PPI (ProtBert) | 70.94 | 73.36 | 70.32 | 78.86 | 67.61 | 87.44 |
| GNN-PPI (OntoProtein)[†] | **72.26** | **78.89** | **75.23** | 77.52 | 76.71 | **91.45** |

Table 2: Protein-Protein Interaction Prediction Results. Breath-First Search (BFS) and Depth-First Search (DFS) are strategies that split the training and testing PPI datasets.

| Method | Transductive BPO | Transductive MFO | Transductive CCO | Inductive BPO | Inductive MFO | Inductive CCO |
|---|---|---|---|---|---|---|
| ProtBert | 0.58 | 0.13 | 8.47 | 0.64 | 0.33 | 9.27 |
| OntoProtein | 0.62 | 0.13 | 8.46 | 0.66 | 0.25 | 8.37 |

Table 3: Protein Function Prediction Results on three sub-sets with two settings. BPO refers to Biological Process, MFO refers to Molecular Function, and CCO refers to Cellular Component.

Note that MSA Transformer takes advantage of multiple sequence alignments (MSAs) and is the current state-of-the-art approach. Finally, we use ProtBert (Elnaggar et al. (2020)) with 30 layers of BERT encoder, which is the largest pre-trained model among baselines.

**Results** We detail the experimental result on TAPE in Table 1. Concretely, we notice that OntoProtein yields better performance in all token level tests. For the second structure (SS-Q3 and SS-Q8) and contact prediction, OntoProtein outperforms TAPE Transformer and ProtBert, showing that it can benefit from those informative biology knowledge graphs in pre-training. Moreover, OntoProtein can achieve comparable performance with MSA transformer. Note that our proposed OntoProtein does not leverage the information from MSAs. However, with external gene ontology knowledge injection, OntoProtein can obtain promising performance. In sequence level tasks, OntoProtein can achieve better performance than ProtBert in fluorescence prediction. However, we observe that OntoProtein does not perform well in protein engineering, homology, and stability prediction, which are all regression tasks. We think this is due to the lack of sequence-level objectives in our pre-training object, and we leave this for future work.

PROTEIN-PROTEIN INTERACTION

**Baselines** We choose four representative methods as baselines for protein-protein interaction. PIPR (Chen et al. (2019)), DNN-PPI (Li et al. (2018)) and DPPI (Hashemifar et al. (2018)) are deep learning based methods. GNN-PPI (Lv et al. (2021)) is a graph neural network based method for better inter-novel-protein interaction prediction. To evaluate our OntoProtein, we replace the initial protein embedding part of GNN-PPI with ProtBERT and OntoProtein as baselines.

**Results** From Table 2, we observe that the performance of OntoProtein is better than PIPR, which demonstrates that external structure knowledge can be beneficial for protein-protein interaction prediction. We also notice th at our method can achieve promising improvement in smaller dataset SHS2K, even outperforming GNN-PPI and GNN-PPI (ProtBert). With a larger size of datasets, OntoProtein can still obtain comparable performance to GNN-PPI and GNN-PPI (ProtBert).

PROTEIN FUNCTION PREDICTION

**Baselines** For simplicity, we leverage Seq2Vec (Littmann et al. (2021)) as the backbone for fair comparison and initialize embeddings with ProtBert and our OntoProtein. Note that our approach is model-agnostic, and other backbones can also be leveraged.

| | 6 ≤ seq < 12 | | | 12 ≤ seq < 24 | | | 24 ≤ seq | | |
|---|---|---|---|---|---|---|---|---|---|
| | P@L | P@L/2 | P@L/5 | P@L | P@L/2 | P@L/5 | P@L | P@L/2 | P@L/5 |
| TAPE Transformer | 0.28 | 0.35 | 0.46 | 0.19 | 0.25 | 0.33 | 0.17 | 0.20 | 0.24 |
| LSTM | 0.26 | 0.36 | 0.49 | 0.20 | 0.26 | 0.34 | 0.20 | 0.23 | 0.27 |
| ResNet | 0.25 | 0.34 | 0.46 | 0.28 | 0.25 | 0.35 | 0.10 | 0.13 | 0.17 |
| ProtBert | 0.30 | 0.40 | 0.52 | 0.27 | 0.35 | 0.47 | 0.20 | 0.26 | 0.34 |
| OntoProtein | **0.37** | **0.46** | **0.57** | **0.32** | **0.40** | **0.50** | **0.24** | **0.31** | **0.39** |

Table 4: Ablation study of contact prediction. $seq$ refers to the sequence length between amino acids. "P@K" is precision for the top $K$ contacts and $L$ is the length of the protein.

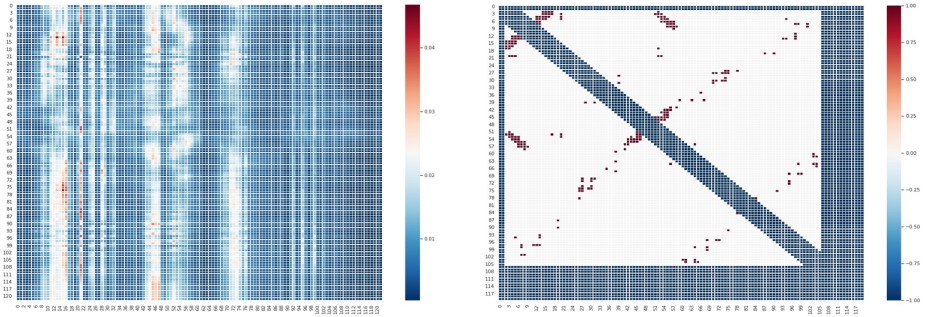

Figure 4: We randomly select a protein from the contact test dataset for visual analysis. **Left**: We visualize the 7th head in the last attention layer in OntoProtein. **Right**: It is the contact label matrix.

**Results** We split the test sets into three subsets (BPO, MFO, and CCO) and evaluate the performance of models separately. From Table 3, we notice that our OntoProtein can yield a 4% improvement with transductive setting and 2% advancement with inductive setting in BPO, further demonstrating the effectiveness of our proposed approach. We also observe that OntoProtein obtain comparable performance in other subsets. Note that there exists a severe long-tail issue in the dataset, and knowledge injecting may affect the representation learning for the head but weaken the tail representation, thus cause performance degradation. We leave this for future works.

## 3.3 ANALYSIS

Table 4 illustrates a detailed experimental analysis on the contact prediction. To further analyze the model's performance, we conduct experiments to probe the performance of different sequences. Specifically, protein sequence lengths from short-range ($6 \leq seq < 12$) to long-range ($24 \leq seq$) are tested with three metrics (P@L, P@L/2, P@L/5). We choose several basic algorithms such as LSTM and TAPE transformer as baselines. For fairness, ProtBert is also leveraged for comparison. It can be seen that the performance of OntoProtein exceeds all other methods in all test settings, which is reasonable because the knowledge injected from Gene Ontology is beneficial. Further, we random sample a protein instance from the test dataset and analyze its attention weight of OntoProtein. We conduct visualization analysis as shown in Figure 4 to compare the contacts among amino acids with the contact label matrix.

## 3.4 DISCUSSION

Applying techniques from NLP to proteins opens new opportunities to extract information from proteins in a self-supervised, data-driven way. Here we show for the first time that injecting external knowledge from gene ontology can help to learn protein representation better, thus, boosting the downstream protein tasks. However, the gains in our proposed OntoProtein compared to previous pre-trained models using large-scale corpus is still relatively small. Note that the knowledge graph ProteinKG25 can only cover a small subset of all proteins, thus, limiting the advancement. We will continue to maintain the knowledge graph by adding new facts from Gene Ontology. Besides, previous studies (Liu et al. (2020); Zhang et al. (2021a)) indicate that not all external knowledge are

beneficial for downstream tasks, and it is necessary to investigate when and how to inject external knowledge into pre-trained models effectively. Finally, our proposed approach can be viewed as jointly pre-training human language and protein (the language of life). Our motivation is to crack the language of life's code with gene knowledge injected protein pre-training. Our work is but a small step in this direction.

# 4 RELATED WORK

## 4.1 PRE-TRAINED LANGUAGE MODELS

Up to now, various efforts have been devoted to exploring large-scale PTMs, either for NLP (Peters et al. (2018); Devlin et al. (2019)), or for CV (Tan & Bansal (2019)). Fine-tuning large-scale PTMs such as ELMo (Peters et al. (2018)), GPT3 (Brown et al. (2020)), BERT (Devlin et al. (2019)), XLNet (Yang et al. (2019)) UniLM (Dong et al. (2019)) for specific AI tasks instead of learning models from scratch has also become a consensus (Han et al. (2021)). Apart from the of large scale language models for natural language processing, there has been considerable interest in developing similar models for proteins (Xiao et al. (2021); Rives et al. (2021)). Rao et al. (2021a) is the first to study protein Transformer language models, demonstrating that information about residue-residue contacts can be recovered from the learned representations by linear projections supervised with protein structures. Vig et al. (2021) performs an extensive analysis of Transformer attention, identifying correspondences to biologically relevant features, and also finds that different layers of the model are responsible for learning different features. Elnaggar et al. (2020) proposes ProtTrans, which explores the limits of up-scaling language models trained on proteins as well as protein sequence databases and compares the effects of auto-regressive and auto-encoding pre-training upon the success of the subsequent supervised training. Human-curated or domain-specific knowledge is essential for downstream tasks, which is extensively studied such as Himmelstein & Baranzini (2015), Smaili et al. (2018), Smaili et al. (2019), Hao et al. (2020), Ioannidis et al. (2020) . However these pre-training methods do not explicitly consider external knowledge like our proposed OntoProtein.

## 4.2 KNOWLEDGE-ENHANCED LANGUAGE MODELS

Background knowledge has been considered as an indispensable part of language understanding ((Zhang et al., 2021a; Deng et al., 2021; Li et al., 2021; Zhang et al., 2019a; Yu et al., 2020; Zhu et al., 2021; Zhang et al., 2021b; Chen et al., 2021; Zhang et al., 2022b; Silvestri et al., 2021; Zhang et al., 2021c; Yao et al., 2022; Zhang et al., 2022a)), which has inspired knowledge-enhanced models including ERNIE (Tsinghua) (Zhang et al. (2019b)), ERNIE (Baidu) (Sun et al. (2019)), KnowBERT (Peters et al. (2019)), WKLM (Xiong et al. (2020)), LUKE (Yamada et al. (2020)), KEPLER (Wang et al. (2021b)), K-BERT (Liu et al. (2020)), K-Adaptor (Wang et al. (2021a)), and CoLAKE (Sun et al. (2020)). ERNIE (Zhang et al. (2019b)) injects relational knowledge into the pre-trained model BERT, which aligns entities from Wikipedia to facts in WikiData. KEPLER (Wang et al. (2021b)) jointly optimizes knowledge embedding and pre-trained language representation (KEPLER), which can not only better integrate factual knowledge into PLMs but also effectively learn KE through the abundant information in the text.

Inspired by these works, we propose OntoProtein that integrates external knowledge graphs into protein pre-training. To the best of our knowledge, we are the first to inject gene ontology knowledge into protein language models.

# 5 CONCLUSION AND FUTURE WORK

In this paper, we take the first step to integrating external factual knowledge from gene ontology into protein language models. We present protein pretraining with gene ontology embedding (OntoProtein), which is the first general framework to integrate external knowledge graphs into protein pre-training. Experimental results on widespread protein tasks demonstrate that efficient knowledge injection helps understand and uncover the grammar of life. Besides, OntoProtein is compatible with the model parameters of lots of pre-trained protein language models, which means that users can directly adopt the available pre-trained parameters on OntoProtein without modifying the ar-

chitecture. These positive results point to future work in (1) improving OntoProtein by injecting more informative knowledge with gene ontology selection; (2) extending this approach to sequence generating tasks for protein design.

## ACKNOWLEDGMENTS

We want to express gratitude to the anonymous reviewers for their hard work and kind comments. This work is funded by NSFCU19B2027/NSFC91846204, National Key R&D Program of China (Funding No.SQ2018YFC000004), Zhejiang Provincial Natural Science Foundation of China (No. LGG22F030011), Ningbo Natural Science Foundation (2021J190), and Yongjiang Talent Introduction Programme (2021A-156-G).

## REPRODUCIBILITY STATEMENT

Our code and datasets are all available in the `https://github.com/zjunlp/OntoProtein` for reproducibility. Hyper-parameters are provided in the Appendix A.3.

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

## A  APPENDIX

### A.1  CONSTRUCTION OF PROTEINKG25

To incorporate Gene Ontology knowledge into language models and train OntoProtein, we construct ProteinKG25, a large-scale KG dataset with aligned descriptions and protein sequences respectively to GO terms and proteins entities. We design two evaluation schemes, the transductive and the inductive settings, which simulate two scenarios of gene annotation in reality.

We use the latest Gene Ontology and Gene Annotations released in April 2020. Gene Ontology depicts the relation between GO terms using GO-GO triplet, and Gene Annotation depicts the relations between protein and GO term using Protein-GO triplet. Due to this connectivity, we combine the two structures into a unified knowledge graph. For each GO term in Gene Ontology, we align it to its corresponding name and description and concatenate them by a colon as an entire description. For each protein in Gene annotation, we align it to the Swiss-Prot and extract its corresponding sequence as its description. The final KG contains 4,990,097 triplets (4,879,951 Protein-GO triplets and 110,146 GO-GO triplets), 612,483 entities (565,254 proteins and 47,229 GO terms) and 31 relations.

Due to the tree-like hierarchical structure of Gene Ontology, we define the depth of GO terms using the shortest path from current GO terms to the root node of GO terms. The distribution of Protein-GO triplets with respect to the depth of GO term is shown at the bottom of Figure 3. Usually, the deeper a GO term is located, the more concrete definition the GO term has, e.g., the small molecule biosynthetic process is the child node of the biosynthetic process. We notice that the number of gene annotations involved with leaf GO terms is a relatively small percentage in all three types of ontologies. We think there are two possibilities: (1) The complexity of annotations of some concrete GO terms, e.g., the identification of whether a protein is involved in the hexose biosynthetic process. (2) The relatively small number of proteins in nature involved with some specific GO terms is intrinsic to these GO terms.

We further pre-process the ProteinKG25 dataset as follows. We observe that the relations of Protein-Go have long-tailed distribution, and mostly focused on *involved_in*, *part_of* and *enables*. Such data distribution will seriously affect the protein feature embedding during pre-training, so we pre-process our dataset to add more precise and fine-grained relations. Figure 5 illustrates the relation distribution of the original model and the distribution after being pre-poccessed. We search for GO terms whose frequency of occurrence in ProteinKG25 is the top 10 in MF and CC, top 20 in BP, then we form a new type of $Protein2GO$ relation by their corresponding relationship such as *parts_of_cytoplasm*.

## A.2 DOWNSTREAM TASK DEFINITION

We list the detailed definition of downstream tasks and its corresponding or similar tasks in nature language processing.

- **Secondary Structure Prediction** is a token-level task and similar to NER (Name Entity Recognition). Each token (amino acid) $x_i$ is mapped to a label $y_i \in \{Helix, Strand, Other\}$.

- **Contact Prediction** is a token-level matching task. Each token (amino acid) pair $x_i$ , $x_j$ of sequence (protein) $x$ is mapped to a label $y_{ij} \in \{0, 1\}$.

- **Remote Homology Detection** is a sequence-level classification task. Each input sequence (protein) $x$ is mapped to a label $y \in \{1, ..., 1195\}$ which represents different possible protein folds.

- **Fluorescence Landscape Prediction** and **Stability Landscape Prediction** are regression tasks where each sequence (protein) $x$ is mapped to a label $y \in \mathbb{R}$ .

- **Protein Protein Interface** is a sequence-level matching task. Each sequence (protein) pair $x_i$ , $x_j$ is mapped to a label $y_{ij} \in \{0, 1\}$.

- **Protein Function Prediction** is a sequence-level classification task or a knowledge graph completion task to prediction link of a protein to the Gene Ontology.

## A.3 EXPERIMENTAL SETTINGS

This section details the training procedures and hyperparameters for each of the datasets. We utilize Pytorch (Paszke et al. (2019)) to conduct experiments with Nvidia V100 GPUs. In pre-training of OntoProtein, similar to Elnaggar et al. (2020), we use the same training protocol such as optimizer, learning rate schedule on BERT model. We set $\gamma$ to 12.0 and the number of negative sampling to 128 in Equation 1.

## A.4 DATAFLOW OF ONTOPROTEIN

We use batches (protein-seq, protein-go, go-go) to jointly train the model, which is shown in Figure 7.

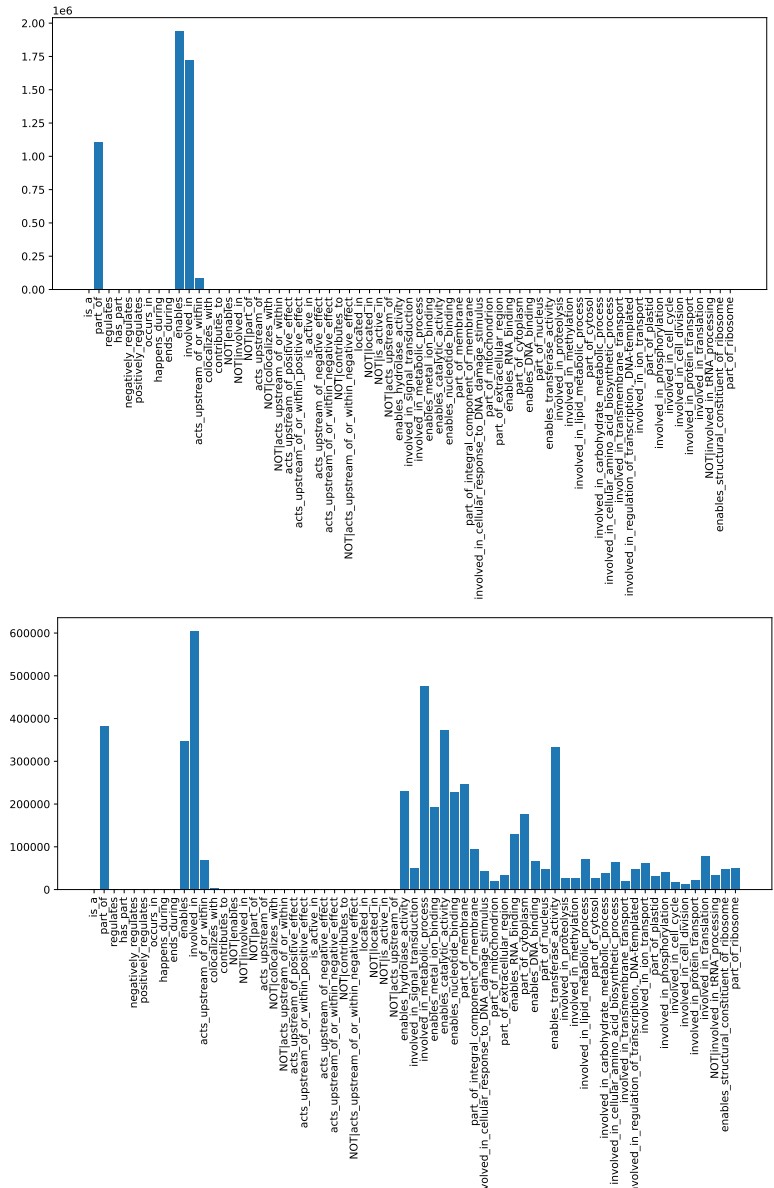

Figure 5: **Top**: Initial relation distribution. **Bottom**: Pre-processed relation distribution.

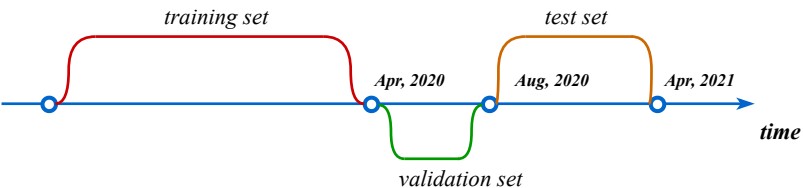

Figure 6: The timeline of the three Gene Annotation datasets. To generate pre-training set and evaluation set of protein function prediction, we choose three Gene Annotation datasets in different periods.

| Entity Type | Relation Type |
|---|---|
| Molecular function | enables, contributes_to |
| Cellular component | located_in, part_of, is_active_in, colocalizes_with |
| Biological process | acts_upstream_of_or_within, involved_in, acts_upstream_of, acts_upstream_of_positive_effect, acts_upstream_of_negative_effect, acts_upstream_of_or_within_positive_effect, acts_upstream_of_or_within_negative_effect |

Table 5: Entity categories of GO terms and specific relation types of these entity categories.

| Task | epoch | batch_size | warmup_ratio | learning_rate | frozen_bert | optimizer |
|---|---|---|---|---|---|---|
| ss3 | 5 | 32 | 0.08 | 3e-5 | False | AdamW |
| ss8 | 5 | 32 | 0.08 | 3e-5 | False | AdamW |
| stability | 5 | 32 | 0.08 | 3e-5 | False | AdamW |
| fluorescence: | 25 | 64 | 0.0 | 3e-5 | True | AdamW |
| remote homology | 10 | 64 | 0.08 | 3e-5 | False | AdamW |
| contact | 10 | 8 | 0.08 | 3e-5 | False | AdamW |

Table 6: Hyper-parameters for the downstream task.

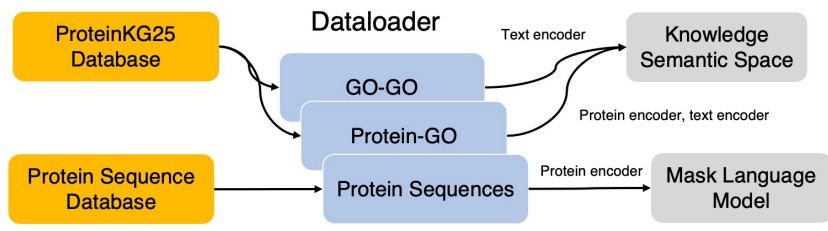

Figure 7: The dataflow of OntoProtein.

| TERM | DESCRIPTION |
|---|---|
| GO annotation | A statement about the function of a particular gene |
| GO term | A standard vocabulary term for biological function annotation |
| GO statement | A specific definition of GO term |

Table 7: Terms descriptions in Gene Ontology. GO annotations are created by associating a gene or gene product with a GO term. Four pieces of information uniquely identify a GO annotation: Gene Product, Go term, Reference, Evidence. And each Go term contains a description text, and this is Go statement.

