# OpenReview forum: "OntoProtein: Protein Pretraining With Gene Ontology Embedding"
_ICLR.cc/2022/Conference — ICLR 2022 Poster_

### Official Review · Reviewer_XNXb · 2021-10-31

**Correctness:** 4
**Technical Novelty And Significance:** 4
**Empirical Novelty And Significance:** 3
**Recommendation:** 6
**Confidence:** 3

**Main Review:**

Good paper but more efforts are needed.

**Summary Of The Paper:**

This paper argues that informative biology knowledge in KGs can enhance protein representation with external knowledge. To show this, the author proposes a computational framework named OntoProtein that makes use of structure in Gene Ontology into protein pre-training models. This paper conducts various experiments to show the superiority of the proposed method and the benefits of the information of Gene Ontology.

**Summary Of The Review:**


pros:
1. In computational biology, the prevailing approaches of learning protein representation rarely consider
incorporating knowledge graphs (KGs), which can provide rich structured knowledge facts. This paper seems to be the first to impose the knowledge graphs of protein.

2. This paper provides a meaningful architecture for simultaneously learning protein knowledge and embedding.

3. To demonstrate the effectiveness of the proposed method, the author constructs a large-scale KG dataset, promoting the research on protein language pre-training.

Cons:

1. To make the paper more friendly for the general ML/DL researchers, I think it should be made clearer the definition of the downstream tasks used in this paper.

2. This paper usually considers one or two baselines in the PPI task, which is are not enough and convincing. It is possible to compare the proposed method with [1] and [2]? If not, please provide the reasons.

3. The presentation of this paper should be further improved. For example, Figures 1 and 4 are hard to read.

[1] Patel, S., Tripathi, R., Kumari, V., & Varadwaj, P. (2017). DeepInteract: deep neural network based protein-protein interaction prediction tool. Current Bioinformatics, 12(6), 551-557.

[2] Liu, X., Luo, Y., Li, P., Song, S., & Peng, J. (2021). Deep geometric representations for modeling effects of mutations on protein-protein binding affinity. PLoS computational biology, 17(8), e1009284.

---

> ### Author Response · Authors · 2021-11-18
> **Response to Reviewer XNXb**
>
> Thank you for the detailed and constructive comments.
>
> - Sorry for the unclear content. We have added the definition of downstream tasks in Appendix A.2. Meanwhile, we correspond the protein tasks with traditional NLP tasks. For example, SSP (Second Structure Prediction) is essentially a NER (Named Entity Recognition) task.
> - We have added baseline results of [1] and [2] for PPI. And all code and datasets are available for reproducibility.
>
> - We have carefully revised Figure 1 and 4.
>
> [1] Hang Li, Xiu-Jun Gong, Hua Yu, and Chang Zhou. Deep neural network-based predictions of protein interactions using primary sequences. Molecules, 23(8):1923, 2018
>
> [2] Maria Littmann, Michael Heinzinger, Christian Dallago, Tobias Olenyi, and Burkhard Rost. Embeddings from deep learning transfer go annotations beyond homology. Scientific reports, 11(1):1–14, 2021.

---

### Official Review · Reviewer_GdWW · 2021-11-03

**Correctness:** 3
**Technical Novelty And Significance:** 2
**Empirical Novelty And Significance:** 2
**Recommendation:** 6
**Confidence:** 4

**Main Review:**

**Strengths**
1. The research direction is important and well motivated -- a lot of valuable information is captured in gene ontologies and ought to be valuable to improve protein representations and performance on downstream task.
2. The paper reads very nicely and easy to follow through, except for a few subsections to clarify / re-write (in particular section 2.4, see below) and would benefit for another read through to correct typos.
3. The method is applied and tested across a diverse set of experiments.

**Weaknesses**
1. Methodological contributions are limited to leveraging existing multi-relational data embedding methods (Bordes et al.) with no adaptation to the specificities of the GO modality. The negative sampling is very close to random sampling (with constraints based on entity group type).
2. This translates into marginal benefits across all experimental setups when comparing with the ProtBert baseline (acknowledged by authors in section 3.4). Some of the claims in section 3.2 do not seem backed up by the results. For example, in the PFP experiments, you cannot simultaneously claim that your method outperforms baselines in BPO and that it performs comparably with other methods in other settings (since baselines outperforms OntoProtein in the latter settings by a larger margin that Ontoprotein outperforms them in the BPO setting).

**Clarifying questions**
1. Section 3.1: “Specifically, we notice that the frequency of leaf GO terms involved in gene annotations and those more specific concepts relative to their parent nodes are less to non-leaf GO terms” -- could you please clarify what you meant here?
2. Section 2.4. Equation 1: you seem to have a typo in equation 1 since it includes r’ but negative triplets involve the same r (r=r’?) since you only negatively sample the leading/tail entities h’ and t’.
3. Section 2.4. Equation 2: why is d not an explicit function of t? Shouldn’t it be d(h+r, t) following Bordes et al?
4. Section 2.4. Equation 3 is confusing. The ‘|’ operator is not standard for sets / not defined anywhere (did you mean union?). Additionally, you state that the negative sampling is achieved by sampling h’ and t’ at random from the same family as h and t resp. -- which I understand means for example that if h is E_{MFO} then also h’ is in E_{MFO} -- but that is not what is expressed mathematically by equation (3). Could you please clarify?
5. Section 2.4: please clarify which embeddings you use exactly here? Are you using aggregate embeddings H_{GO} and H_{Protein} and none of the token level embeddings?
6. Section 2.4: Do you do any negative sampling for T_{protein--GO} triplets? Are you limiting to negatively sampling tail (GO) entities?
7. Section 2.5: How do you form input batches to jointly train this objective?
8. Figure 4: How is it different from what one would obtain with ProtBert?

**Minor points**
1. Introduction: "different from knowledge-enhanced approaches in NLP" → please cite
2. Introduction: “text deceptions” (towards the top of page 2) -- did you mean "test descriptions"?
3. Would suggest that you give 1 or 2 concrete examples to illustrate the different objects described in 2.2 and appendix A.1. Table 7 is also a bit confusing (e.g. difference between GO term Vs GO statement?)
4. The write up of section 2 could be improved -- there was a lot of overlap between section 2.1., 2.2 and 2.3 which was making some information redundant / making things more difficult to read
5. Section 2: would suggest to include an overall architecture diagram to show how an example input triplet is processed (eg., what goes to which encoder type?).
6. Section 3: bolding of results is not consistent / misleading. Always bold the best result in each column.
7. Section 3: would be helpful to have MSA transformer everywhere to put things in perspective (bearing in mind it does have access to MSA data that you do not explicitly leverage).


**Summary Of The Paper:**

This paper introduces a method to enrich the representations that are learnt by protein language models with knowledge encapsulated in gene ontologies. To do so, it curates a knowledge graph (ProteinKG25) and applies existing methods in multi-relational data embedding (Bordes et al.) to jointly train knowledge embeddings and protein embeddings, with the objective to enhance the latter and subsequently improve the performance on various downstream tasks (namely TAPE benchmark, protein-protein interaction and protein function prediction).

**Summary Of The Review:**

The research direction of this paper is very important and that the work described here is adequately framed and motivated. A valuable contribution is the ProteinKG25 knowledge graph put together by authors, which could be valuable to other researchers. While there are several points to clear out (see main review), the ideas are overall well presented and the paper reads nicely.
However, the methodological and empirical contributions of this work are somewhat limited: the Knowledge Embedding for the knowledge graph (while applied for the first time to GO data) is largely borrowed from Bordes et al. and not particularly tailored to this very specific data modality; the negative sampling is only marginally different from random sampling (constraining on entity group). To the author’s own account “the gains in our proposed OntoProtein compared to previous pre-trained models using large-scale corpus is still relatively small” since “the knowledge graph ProteinKG25 can only cover a small subset of all proteins, thus, limiting the advancement”. In my view, this is precisely the main challenge to resolve for this type of work attempting to instill established knowledge into large protein language models, which would have made this work a very meaningful contribution.
All things considered, I’m leaning weak reject given the minor methodological contributions and weak experimental results.

---

> ### Author Response · Authors · 2021-11-18
> **Response to Reviewer GdWW**
>
> We would like to express our great appreciation to reviewers for their comments on our paper. We highlight that an in-depth understanding of OntoProtein is crucial for understanding our contributions. Specifically, the importance of our model can be summarized from three dimensions: (a) from an engineering perspective, it directly provides a solution about how to inject gene ontology knowledge into protein pre-training (b) from an educational perspective, the empirically results helps us to better understand the how knowledge help protein understanding (c) from a theoretical perspective, it serves as a scaffold for leveraging heterogeneous knowledge (graph, text, proteins) into a unified pre-training framework.
>
> We note that as we aim to make our idea easy to follow, we introduce the OntoProtein in a slow, step-by-step fashion. Technically, the heterogeneity of knowledge or multi-modal fusion (protein, text, graph) is an open challenge. We try to address this by integrating heterogeneous information into the same semantic space. We further design a hybrid encoder to represent language text and protein sequence and introduce contrastive learning with knowledge-aware negative sampling to jointly optimize the knowledge graph and the protein sequence embedding during pre-training. However, it may not be that straightforward to develop such modules if not in our context, which is a crucial component in heterogeneity knowledge fusion. Moreover, we release all the source code and a large-scale knowledge graph ProteinKG25 for the community, which may attract researchers from both biology and computer science.
>
> For experiments, in most of the test cases, OntoProtein outperforms ProtBert; however, it cannot yield better performance in a few tasks. Due to resource limitations, we cannot tune the hyperparameters, train more steps, and leverage larger knowledge graphs that may cover more proteins. We think with careful hyperparameter tuning and more pre-training steps, and we can obtain better performance. Overall, our approach provides a proof of concept that injecting ontology knowledge into protein representation learning is beneficial.
>
> For negative sampling, we think OntoProtein carefully considers the adaption to GO structure. If we conduct randomly negative sampling, we may get some easy or meaningless negative samples. Thus, we propose knowledge-aware negative sampling, which replaces GO entities according to their related group type. And we use specific strategies for different types of triples. For GO-GO type triple, we replace head or tail entities to get corrupted triples, but for GO-protein type triple, we only replace protein entities to get meaningful negative samples. We agree with your concerns that negative sampling is an important procedure. We will investigate some more sophisticated methods such as sequence alignment and homology search to obtain better negative samples in the future.
>
> We have revised all the mistakes (marked in red) in Sections 2.4 and 2.5.
>
> Other minor points to clarify:
>
> - Sorry for the confusing parts and mistakes in Sections 2.4 and 2.5; we have carefully revised the mistakes. The detailed explanations of leaf and non-leaf GO teams are in Appendix A1.
>
> - Sorry for the typo. We have revised Equation 1, Equation 2, and Equation 3.
>
> - The embedding of the GO node is obtained by the GO (text) encoder, and the embedding of the protein node is obtained by the protein encoder (shared by mask language model). In the implementation, we use mean-pooling and an extra linear layer to aggregate token embeddings.
>
> - We do negative sampling for T_{protein--GO} triplets, and we limit to only sample tail (GO) entities.
>
> - We use batches (protein-seq, protein-go, go-go) to jointly train the model (seen in Figure 7); detailed implementation in codes can be found in the Supplementary Materials.
>
> - For Table 4, we conduct ablation studies to show OntoProtein performs better than ProtBert in contacts prediction and visualize the attention layers of OntoProtein to analyze its performance in Figure 4. Since attention visualization is not a
> a quantifiable way for comparison, so we list the detailed results in Table 4.
>
> - We have added the missing references and typos.
>
> - We have added examples to illustrate the difference between the GO term and GO statement in Table 7.
>
> - We have carefully revised Section 2.
>
> - We have added an overall architecture diagram to show how an example input triplet is processed in Figure 7 (Appendix).
>
> - We have carefully checked the bolding of the results.
>
> - Good suggestion. We think it would be helpful. Because re-training MSA transformers and fine-tuning all downstream tasks require huge computation resources. We will try to utilize MSA transformer as a backbone in the future.

---

> > ### Comment · Reviewer_GdWW · 2021-11-30
> > **Final thoughts**
> >
> > Dear authors,
> > Thank you very much for your thorough response and all improvements made to the paper during rebuttal (note: minor issue with bolding in Table 1 remains). I am raising my score accordingly. I do think the main idea from this work (using gene ontologies to enrich representations learnt by protein language models) has a lot of potential, that the paper is well written and the approach is sensible. I am not recommending acceptance more strongly as the main points discussed in "Summary Of The Review" remain post rebuttal.

---

### Official Review · Reviewer_fXpi · 2021-11-03

**Correctness:** 3
**Technical Novelty And Significance:** 3
**Empirical Novelty And Significance:** 3
**Recommendation:** 6
**Confidence:** 4

**Details Of Ethics Concerns:**

Not applicable.

**Main Review:**

Reasons to accept:
* Interdisciplinary study on extending the language model and KG embedding into protein embedding to empower bio appications
* Insightful strategy and observation on KG negative sampling
* Extensive experiments on multiple tasks and ablation study (attention analysis) are conducted. New useful resources of enhanced protein datasets are created and available publicly with strong reproducibility.

Reasons to reject:
* The information fusion of text token and protein sequence tokens may face modality challenges in one KG embedding model.
* Some missing references and potential baselines on the PPI task and other works that jointly trained on protein and gene ontology (see more in detailed comments)
* Lack of quality assessment of curated dataset ProteinKG25.

Detailed comments:

* One of the major concerns of this paper is that some existing works with joint training on protein with other domains. Examples are [1,2,3] with PPI predictions as one task for evaluation (can be added to PPI baselines). As for PPI baselines, since the sequences have been added as input, the current evaluation only considered GNN-PPI variations, however, there is a large collection of existing work relying on protein sequences (potentially with the help of PSI-BLAST and multiple sequence alignment) and follow-up works of PIPR. It is necessary to show that the purely sequence-based model cannot compete with the proposed OntoProtein with an additional facet of GO knowledge.
* In OntoProtein, the GO entity embeddings are based on text description while proteins are encoded with their amino acid sequence tokens through a shared model in MPM. The features from different modalities are jointly optimized together through KE loss with TransE. Two features are computed in the same embedding space with a simple translational distance which does not seem to make much sense, especially there are two types of triples (Protein-GO and GO-GO) trained together. This requires more justification on what the embeddings aim to learn.
* It seems that the performance of OntoProtein does not significantly outperform other baselines in most cases on the TAPE benchmark. What is the benefit of OntoProtein compared with other models (especially ProtBERT)?
*  It is interesting to see how different aspects can be leveraged into better protein presentations. Since the paper claims that the protein representation learning benefits from gene ontology, and GO can be divided into BP/MF/CO components, it would be interesting to see how individual subsets in GO can result in different improvements. Also, it is said that gene ontology (three subsets) and PPI are not independent of each other (some protein functions are annotated due to their known binding activities with other proteins). It is suggested to see whether such correlations can be identified in the current training framework or the OntoProtein can help solve the incompleteness.
* There is no quality assessment of the newly curated dataset ProteinKG25. The understanding is that the authors successfully made one collection aligned with external resources of annotations to generate the protein-centric KG. But it is unclear about the quality (accuracy and completeness). It is also suggested to make a comparison of other benchmarks such as HetioNet[4] and DRKG[5] in terms of the detailed information or KG schema.
* No sensitivity study on the hyperparameter \alpha in pre-training objective which balances MLM and KE losses.

Minor suggestions and corrections:
* Figure 1 is somewhat confusing about the nodes and edges, especially the left subfigure and the number annotations. It would be beneficial to layout one snapshot of created Protein
* The acronym “MLM” (masked language model) was used first in the introduction without full name (Page 2 line 10).

References:
* [1] Onto2Vec: Smaili, F. Z., Gao, X., & Hoehndorf, R. (2018). Onto2vec: joint vector-based representation of biological entities and their ontology-based annotations. Bioinformatics, 34(13), i52-i60.
* [2] OPA2Vec: Smaili, F. Z., Gao, X., & Hoehndorf, R. (2019). Opa2vec: combining formal and informal content of biomedical ontologies to improve similarity-based prediction. Bioinformatics, 35(12), 2133-2140.
* [3] Bio-JOIE: Hao, J., Ju, C. J. T., Chen, M., Sun, Y., Zaniolo, C., & Wang, W. (2020, September). Bio-JOIE: Joint representation learning of biological knowledge bases. In Proceedings of the 11th ACM International Conference on Bioinformatics, Computational Biology and Health Informatics (pp. 1-10).
* [4] HetioNet: Himmelstein, D. S., & Baranzini, S. E. (2015). Heterogeneous network edge prediction: a data integration approach to prioritize disease-associated genes. PLoS computational biology, 11(7), e1004259.
* [5] DRKG: Ioannidis, Vassilis N., Song, Xiang, Manchanda, Saurav, Li, Mufei, Pan, Xiaoqin, Zheng, Da, Ning, Xia, Zeng, Xiangxiang & Karypis, George. (2020) Drug Repurposing Knowledge Graph for Covid-19. https://github.com/gnn4dr/DRKG/


**Summary Of The Paper:**

This paper introduces OntoProtein a comprehensive pre-training framework for protein embedding with the knowledge of gene ontology (GO). More specifically, OntoProtein jointly optimizes on both masked Protein Model and Knowledge Graph Embedding model which results in knowledge-aware protein embedding for downstream applications including protein-protein interactions and protein GO association prediction. The authors also create a new benchmark of proteins with aligned annotations to facilitate the OntoProtein training.

**Summary Of The Review:**

This paper has merits in multiple directions. It applies the state-of-the-art language modeling in NLP in protein representation learning named OntoProtein. Also, OntoProtein help creates a new augmented protein benchmark with aligned GO annotations (enhanced views for heterogenous BioKG with multiple domains). The knowledge-aware negative sampling strategy has underlying insights on utilizing the hierarchically structured gene ontology which can be generalized onto KG negative sampling with ontology information, not limited in bio domain. The critics of this work are mostly about some missing references and baseline approaches, as well as the limitation on combinations of existing works.  Overall, the reviewer agrees that the merits of this work outweigh the flaws.

---

> ### Author Response · Authors · 2021-11-18
> **Response to Reviewer fXpi**
>
> Thank you for the detailed and constructive comments.
>
> - We have added baseline results for PPI. Our proposed OntoProtein is designed as a general pre-training framework with the help of gene ontology. As the training corpus (protein sequences) is huge and abundant, we can use our pre-trained protein encoder to encode seen or unseen protein in downstream tasks, which will be more efficient and straightforward than PSI-BLAST or multiple sequence alignment.
>
>     To make a fair comparison, we try not to include task-specific information such as protein networks in PPI. We agree with our suggestion in the fourth detailed comment; some protein functions are annotated due to their known binding activities with other proteins, it is beneficial to leverage such a mechanism, and we will leave this for future works.
>
> - Sorry for the confusing points. We have revised this part. During the pre-training, we use extra linear layers to project different modalities of text description and protein sequences into the same semantic space, and the protein and text encoders are different. Actually, there exist challenges of modality fusion in the fusion process, and it is worthwhile to further investigate which part of knowledge plays a more crucial role in the pre-training. We will discuss and analyze the problem of modality fusion in future works.
>
> - OntoProtein yields better performance in all classification tasks. However, we observe that OntoProtein does not perform well in protein engineering, homology, and stability prediction, which are all regression tasks.
>
> - In the protein function prediction task, we have tested the performance in transductive and inductive settings in BP/MF/CO components in Table 3.
>
> - We use the latest Gene Ontology and Gene Annotations released in April 2020 to construct ProteinKG25. Note that the Gene Ontology (GO) is a **high-quality knowledge base** which is the world’s largest source of information on the functions of genes. This knowledge is both human-readable and machine-readable and is a foundation for computational analysis of large-scale molecular biology and genetics experiments in biomedical research. We will conduct crowdsourcing to further evaluate the quality of the ProteinKG25.
>
> - It is difficult to conduct sensitivity analysis about hyper-parameter \alpha because of the computation-intensive pre-training process (about several days with Nvidia V100), and we will leave this for future works.

---

### Author Response · Authors · 2021-11-18
**Summary of Revisions**

Dear reviewers and AC,

We sincerely appreciate your valuable time and constructive comments.

We’ve uploaded a revised draft incorporating reviewer feedback. Below is a summary of the main changes:

- Figure 1 with concrete examples
- Figure 4 for clear explanations.
- Add baseline results in PPI tasks.
- Add the pre-processing procedure about ProteinKG25 in Appendix A.1 and Figure 5.
- Add the definition of downstream tasks in Appendix A.2.
- Correct mistakes in Section 2.4, 2.5
- Add missing references in Section 3, 4
- Add dataflow of OntoProtein in Figure 7.

We briefly introduce the motivation, method, and contribution as follows:

Motivation: Integrate external knowledge graphs (gene ontology) into protein pre-training.

Method:
- Hybrid encoder to represent language text and protein sequence
- Contrastive learning with knowledge-aware negative sampling to jointly optimize the knowledge graph and the protein sequence embedding.

Contribution:

- The first knowledge-enhanced protein pre-training approach that brings promising improvements to a wide range of protein tasks.
- By contrastive learning with knowledge-aware sampling to jointly optimize knowledge and protein embedding, OntoProtein shows its effectiveness in widespread downstream tasks, including protein function prediction, protein-protein interaction prediction, contact prediction, and so on.
- We construct and release the ProteinKG25, a novel large-scale KG dataset, promoting the research on protein language pre-training.
- Code and datasets are in the supplementary material and will be released for reproducibility.

We sincerely hope our responses and revisions address all reviewers’ concerns.

We sincerely believe that these updates may help us better deliver the benefits of the proposed OntoProtein to the ICLR community.

Thank you very much,

Authors.

---

### Decision · Program_Chairs · 2022-01-20

**Decision:**

Accept (Poster)

**Comment:**

This paper introduce a protein pretraining framework that enhances representations learnt from protein language modeling with knowledge graph embeddings. The new framework, OntoProtein, optimizes jointly a masked Protein objective and a Knowledge Graph Embedding objective producing knowledge-aware protein embeddings. These embeddings are evaluated on downstream tasks including protein-protein interactions and protein GO association prediction. The paper also introduces a new large-scale KG dataset, ProteinKG25.

The reviewers were in agreement that the paper presents an important research direction and that the work is well framed and motivated. The dataset contribution was also considered important by the reviewers and they are in agreement that the paper is clear and generally easy to understand. Reviewers were concerned that the novelty of the work is in the application of existing techniques to a new domain rather than introducing new domains, but generally the reviewers considered the novelty to be sufficient for publication. There were some other concerns about missing references and some of the presentation, but the authors addressed these concerns in their response and in the updated version that they produced.